EMBO
Molecular Medicine

# miR-29 contributes to normal endothelial function and can restore it in cardiometabolic disorders

Michael E Widlansky[1],*,†  [ID], David M Jensen[2],†, Jingli Wang[1], Yong Liu[2], Aron M Geurts[2], Alison J Kriegel[2], Pengyuan Liu[2], Rong Ying[1], Guangyuan Zhang[2], Marc Casati[2], Chen Chu[2], Mobin Malik[1], Amberly Branum[1], Michael J Tanner[1], Sudhi Tyagi[1], Kristie Usa[2] & Mingyu Liang[2],** [ID]

## Abstract

We investigated the role of microRNAs (miRNA) in endothelial dysfunction in the setting of cardiometabolic disorders represented by type 2 diabetes mellitus (T2DM). miR-29 was dysregulated in resistance arterioles obtained by biopsy in T2DM patients. Intraluminal delivery of miR-29a-3p or miR-29b-3p mimics restored normal endothelium-dependent vasodilation (EDVD) in T2DM arterioles that otherwise exhibited impaired EDVD. Intraluminal delivery of anti-miR-29b-3p in arterioles from non-DM human subjects or rats or targeted mutation of *Mir29b-1/a* gene in rats led to impaired EDVD and exacerbation of hypertension in the rats. miR-29b-3p mimic increased, while anti-miR-29b-3p or *Mir29b-1/a* gene mutation decreased, nitric oxide levels in arterioles. The mutation of *Mir29b-1/a* gene led to preferential differential expression of genes related to nitric oxide including Lypla1. Lypla1 was a direct target of miR-29 and could abrogate the effect of miR-29 in promoting nitric oxide production. Treatment with Lypla1 siRNA improved EDVD in arterioles obtained from T2DM patients or *Mir29b-1/a* mutant rats or treated with anti-miR-29b-3p. These findings indicate miR-29 is required for normal endothelial function in humans and animal models and has therapeutic potential for cardiometabolic disorders.

**Keywords** diabetes; endothelium; hypertension; microRNA; nitric oxide

**Subject Categories** Metabolism; Vascular Biology & Angiogenesis

## Introduction

Microvascular complications of diabetes mellitus (DM) are a leading or major cause of renal failure, blindness, and diabetes-related cardiomyopathy. The cascade of events leading to DM microvascular disease begins with the development of vascular endothelial dysfunction (Nitenberg *et al*, 1993; Widlansky *et al*, 2003). Endothelial dysfunction describes the readily detectable pro-thrombotic, pro-inflammatory, pro-vasoconstrictive, and pro-proliferative phenotype that predicts future adverse vascular-related events and precedes the development of clinical disease (Widlansky *et al*, 2003). Endothelial dysfunction is a common finding in DM and associated diseases including hypertension (Dharmashankar & Widlansky, 2010).

MicroRNAs (miRNAs) play an important role in the regulation of protein expression (Ambros, 2004; Bartel, 2004; Couzin, 2008; Makeyev & Maniatis, 2008; Krol *et al*, 2010). Their structure enables single miRNAs to have coordinated effects on multiple targets and pathways, which may then converge to significantly impact a functional process (Liang, 2009; Boon *et al*, 2012; Neth *et al*, 2013). Several studies have identified miRNA signatures in the plasma and circulating leukocytes of patients with DM (Zampetaki *et al*, 2010; Kong *et al*, 2011; Karolina *et al*, 2012; Ortega *et al*, 2014), and miRNAs have been implicated in a wide array of diabetic complications (McClelland & Kantharidis, 2014). However, circulating miRNA signatures might not reflect tissue level expression. Whether miRNA expression patterns are altered in the microvasculature of DM patients remains unknown. Several miRNAs have been shown to influence some aspects of endothelial function (Anand *et al*, 2010; Sun *et al*, 2012; Schober *et al*, 2014; Vikram *et al*, 2016). However, it remains unknown whether miRNAs can influence the development of endothelial dysfunction in human cardiovascular diseases including microvascular complications of DM.

To address these knowledge gaps, we obtained human resistance arterioles, vessels directly relevant to microvascular disease, from patients at high cardiometabolic risk for microvascular disease and non-DM controls through biopsy, and performed endothelial function and miRNA expression analysis. We referred to the patients involved in the current study as type 2 DM (T2DM) for simplicity but recognized that T2DM is frequently associated with hypertension, dyslipidemia, and obesity. The focus of the current study was on endothelial dysfunction

1  Division of Cardiovascular Medicine, Department of Medicine, Medical College of Wisconsin, Milwaukee, WI, USA
2  Department of Physiology, Center of Systems Molecular Medicine, Medical College of Wisconsin, Milwaukee, WI, USA
   *Corresponding author. Tel: +1 (414) 955 6755; Fax: +1 (414) 955 6203; E-mail: mwidlans@mcw.edu
   **Corresponding author. Tel: +1 (414) 955 8539; E-mail: mliang@mcw.edu
   †These authors contributed equally to this work

occurring in patients with this clinically highly relevant constellation of cardiometabolic disorders. Subsequent studies in human resistance arterioles, miRNA gene mutant rats, and cultured endothelial cells indicated that miR-29 is required for normal endothelial function and can restore endothelium-dependent vasodilation in resistance arterioles from T2DM patients by promoting nitric oxide (NO) production. The mechanism involves suppression of expression of lysophospholipase I (Lypla1), a protein critical to depalmitoylation of eNOS and its detachment from plasma membrane caveolae.

# Results

### Endothelial dysfunction and altered miRNA expression in human T2DM arterioles

We recruited an initial cohort of 18 subjects with T2DM and 20 non-DM control subjects for analysis of vascular endothelial function and arteriolar miRNA expression. Demographic, clinical, and medication information for this initial cohort is available in Appendix Table S1. *In vivo* measurements from brachial artery reactivity testing indicated resting flow, a reflection of microvascular function, was significantly lower in T2DM subjects ($63 \pm 4$ vs. $90 \pm 11$ ml/min, $P = 0.035$, Student's *t*-test; Appendix Table S1). Absolute and percent flow-mediated dilation and percent nitroglycerin-mediated vasodilation tended to be lower in T2DM subjects but did not reach statistical significance (Appendix Table S1).

Endothelial function in resistance arterioles was examined *ex vivo* in gluteal adipose arterioles isolated from a subset of these subjects ($n = 7$ for T2DM and $n = 9$ for non-DM controls). Acetylcholine (Ach)-induced vasodilation was significantly impaired in T2DM subjects compared to controls based on both the vasodilatory response to peak-dose Ach ($44 \pm 9\%$ vs. $69 \pm 6\%$, $P = 0.04$, Student's *t*-test) and by analyses of the entire Ach dose–response curve ($P < 0.001$ by 2-way ANOVA by DM status). No differences were seen in endothelium-independent vasodilation or smooth muscle reactivity as tested by response to 0.2 mM papaverine ($100 \pm 1\%$ vs. $99 \pm 2\%$, $P = 0.16$).

Small RNA deep sequencing was performed in gluteal adipose arterioles obtained from the entire groups of subjects in this initial cohort ($n = 18$ for T2DM and $n = 20$ for non-DM; Dataset EV1). Overall, 71 of 763 detected miRNAs showed unadjusted $P$-values less than 0.05 when DM arterioles were compared to non-DM controls, with FDR ranging from 0.14 to 0.53. The top 30 miRNAs showed unadjusted $P$-values less than 0.015 and FDR less than 0.4. Of the top 30 miRNAs, 18 are known human miRNAs (Appendix Table S2). Several of these 18 miRNAs have some known involvement in vascular function or have been shown to be present at different abundance levels in the plasma of diabetic subjects compared to non-diabetic subjects (described further in Discussion section and in bold in Appendix Table S2).

Three of the 18 top miRNAs, hsa-miR-29b-3p ($P = 0.015$), hsa-miR-29a-3p ($P = 0.005$), and hsa-miR-146b-5p ($P = 0.003$) were further analyzed in the original arteriole RNA samples using real-time PCR. The real-time PCR result (Appendix Fig S1) confirmed significant up-regulation of these miRNAs in DM arterioles as indicated by the deep sequencing analysis.

### miR-29 restores endothelium-dependent vasodilation in human T2DM arterioles

Notably, several members of the miR-29 family were among the 18 top miRNAs (Appendix Table S2 and Fig 1A). The up-regulation of miR-29 family members in T2DM arterioles suggested miR-29 could either contribute to the endothelial dysfunction in T2DM or was up-regulated in a failed attempt to compensate for the endothelial dysfunction. These findings prompted us to further investigate the impact of molecular manipulation of miR-29 on human microvascular function through recruitment of a second cohort of subjects. Demographic and clinical profiles of subjects involved in miR-29 manipulation experiments were similar to the subjects involved in the miRNA deep sequencing analysis (Appendix Table S3). We treated human T2DM arterioles with LNA-modified anti-miR-29b-3p oligonucleotide delivered directly into the lumen of the arterioles. Intraluminal treatment with anti-miR-29b-3p did not improve acetylcholine-induced vasodilation (Fig 1B), suggesting the up-regulation of miR-29b in T2DM arterioles did not contribute to the endothelial dysfunction in these arterioles.

Intraluminal transfection with miR-29b-3p or miR-29a-3p mimics, on the other hand, significantly improved acetylcholine-induced vasodilation in T2DM arterioles (Fig 1C and D). Papaverine-induced dilation was not affected (miR-29b mimic: $95 \pm 3\%$ for control mimic vs. $96 \pm 1\%$ for miR-29b mimic, $N = 9$, $P = 0.66$, Student's *t*-test; miR-29a mimic: $79 \pm 11\%$ for control mimic vs. $90 \pm 5\%$ for miR-29a mimic, $N = 5$, $P = 0.31$, Student's *t*-test). The transfection efficiency was confirmed by real-time PCR analysis showing accumulation of miR-29 beyond the already high levels in T2DM arterioles (Appendix Fig S2). The vessels transfected with scrambled oligonucleotides dilated to approximately 20–30% of maximum diameter in response to the peak dose of Ach in the experiments shown in Fig 1B and D, which was lower than untransfected DM vessels described above (dilated to 44%). The difference was likely because the experiment in untransfected DM vessels was performed in vessels immediately after isolation in physiological buffer, while the experiments in Fig 1B and D were performed in vessels that had been transfected for 4 h and perfused overnight (~20 h) in a myograph in cell culture conditions. As shown later in Fig 2A, the peak Ach-induced dilation of non-DM vessels after similar transfection and overnight perfusion would be approximately 50–60% of maximum dilation. Therefore, the levels of endothelium-dependent vasodilation in T2DM arterioles transfected with miR-29 mimics were similar to the levels in non-DM arterioles under comparable experimental settings, indicating treatment with miR-29 mimics restored endothelium-dependent vasodilation in T2DM arterioles.

These data support the hypothesis that the up-regulation of miR-29 in human T2DM arterioles was a failed attempt to compensate for the endothelial dysfunction. It appears possible that the endogenously up-regulated miR-29 may be rendered nonfunctional in T2DM arterioles, the mechanism for which remains to be investigated. It is clear, however, that further elevation of miR-29 levels by intraluminal transfection with miR-29 mimics could restore normal endothelial function in human T2DM arterioles.

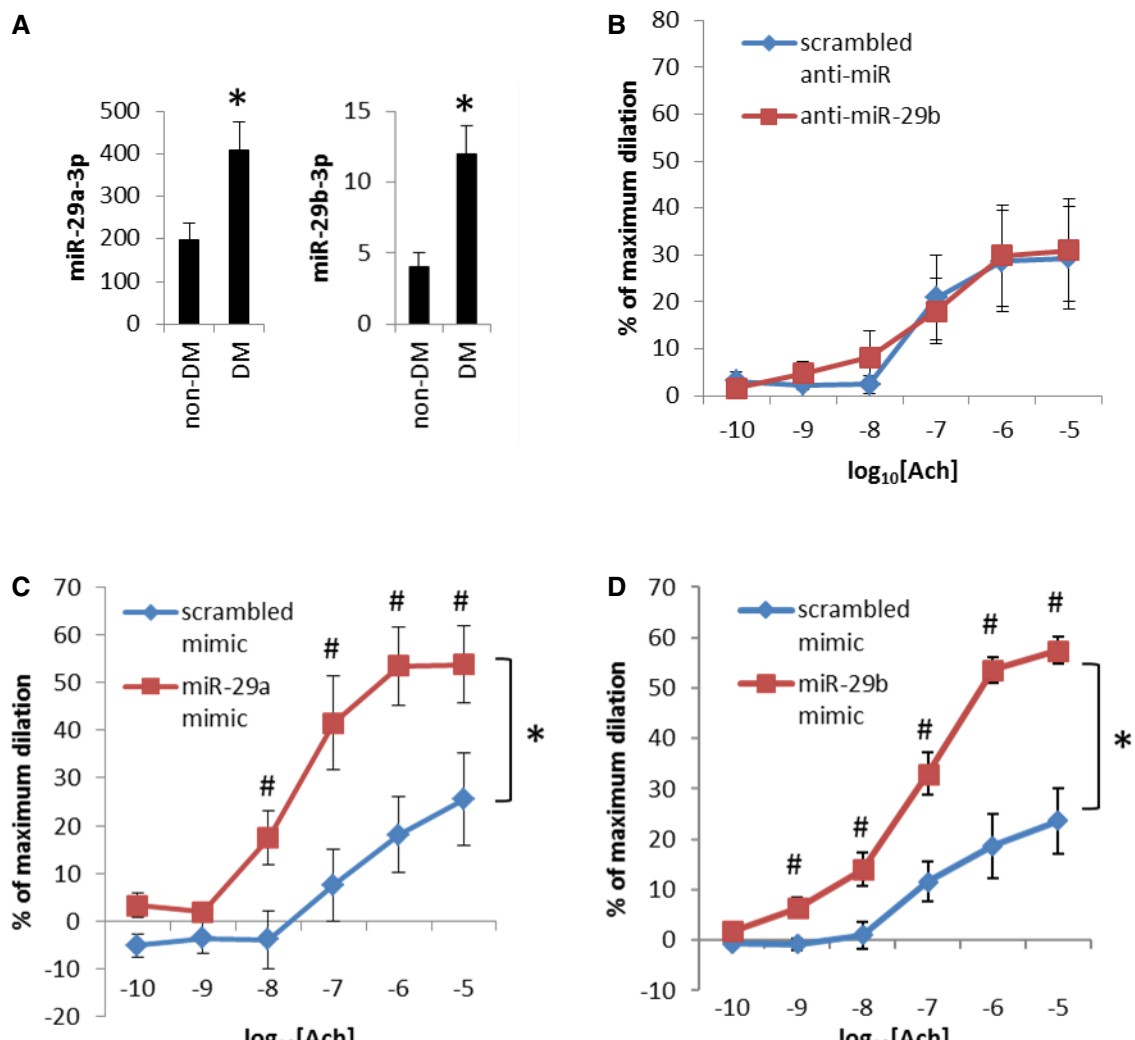

**Figure 1.  miR-29 restores endothelium-dependent vasodilation in resistance arterioles obtained from T2DM patients.**

A  Alteration of miR-29a-3p and miR-29b-3p abundance in T2DM arterioles. Data were obtained from small RNA deep sequencing. $N$ = 20 non-DM and 18 DM, *$P$ < 0.05.

B  Intraluminal transfection with anti-miR-29b-3p did not improve Ach-induced vasodilation in arterioles from T2DM patients. $N$ = 5. The levels of Ach-induced vasodilation in these vessels were comparable to the levels seen in non-DM vessels treated with anti-miR-29b-3p shown later in Fig 2A.

C  Intraluminal transfection with miR-29a-3p mimic restored normal Ach-induced vasodilation in arterioles from T2DM patients. $N$ = 5, *$P$ < 0.05, two-way ANOVA; #$P$ < 0.05 at the given concentration of Ach, Holm–Sidak test.

D  Intraluminal transfection with miR-29b-3p mimic improved Ach-induced vasodilation in arterioles from T2DM patients. $N$ = 9, *$P$ < 0.05, two-way ANOVA; #$P$ < 0.05 at the given concentration of Ach, Holm–Sidak test.

Data information: Data are reported as mean ± SEM.

## miR-29 is required for normal endothelial function in human and an animal model

The strong corrective effect of miR-29 on endothelial dysfunction in T2DM arterioles prompted us to further investigate the role of miR-29 in normal endothelial function and the mechanisms involved. We treated arterioles obtained from non-diabetic subjects, which exhibited normal endothelium-dependent vasodilation, with LNA-modified anti-miR-29b-3p oligonucleotide delivered intraluminally. Intraluminal treatment with anti-miR-29b-3p substantially impaired acetylcholine-induced vasodilation (Fig 2A), reducing the peak Ach response by 58% ($N$ = 8, $P$ = 0.005). Papaverine-induced dilation

was not affected (95 ± 2% for scrambled control vs. 96 ± 2% for anti-miR-29b-3p, $P$ = 0.87, Student's $t$-test), and spermine NONOate, a nitric oxide (NO) donor, induced robust dilation in arterioles after intraluminal treatment with anti-miR-29b-3p (Appendix Fig S3), indicating the intraluminal anti-miR-29b-3p treatment did not directly influence the smooth muscle's ability to undergo NO-induced relaxation. Similar to non-diabetic arterioles, intraluminal treatment with anti-miR-29a-3p or anti-miR-29b-3p significantly impaired acetylcholine-induced vasodilation in gluteal arterioles isolated from rats (Fig 2B). These data indicate that miR-29 is required for normal endothelial function in human and rat arterioles.

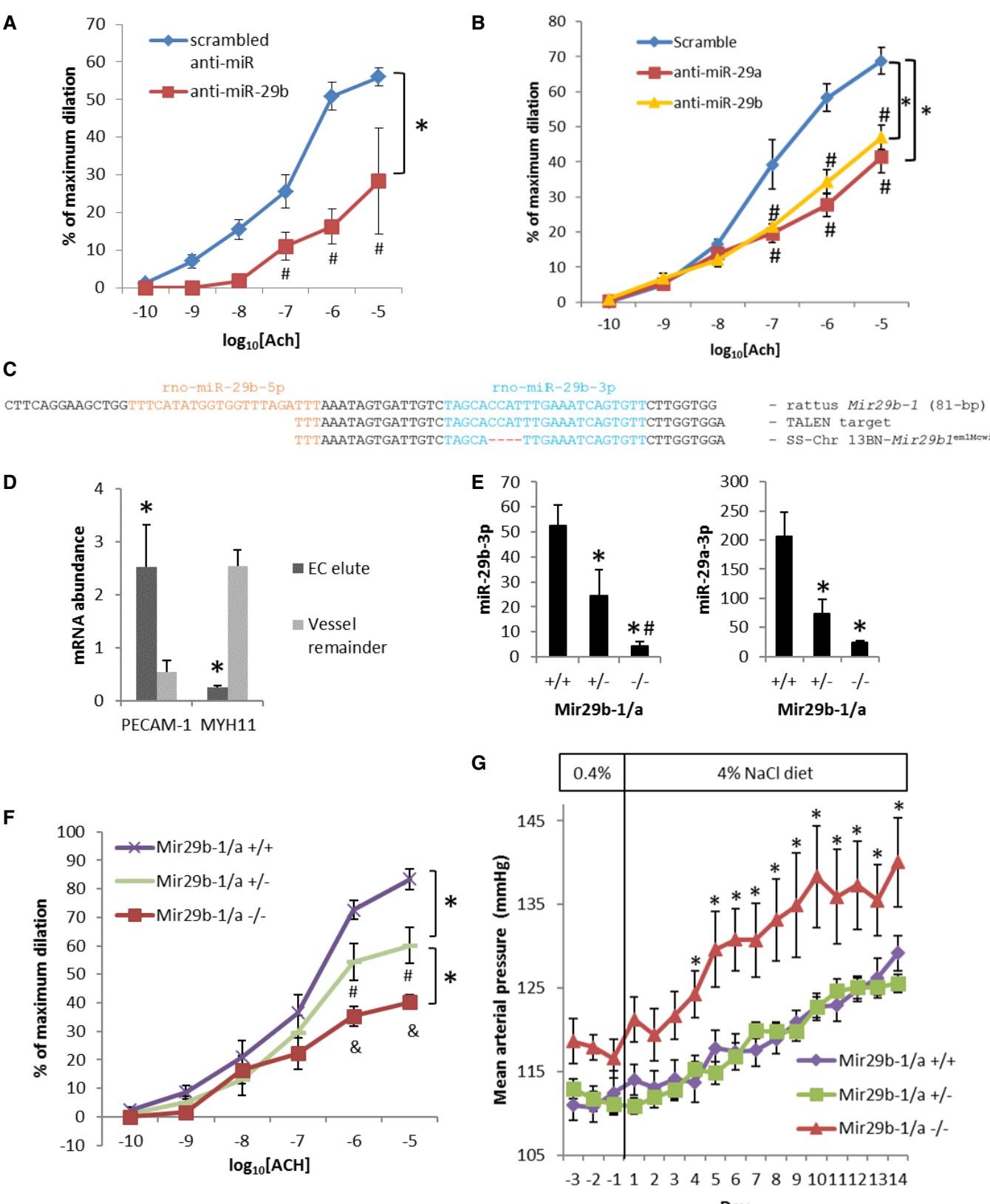

Figure 2.

**Figure 2.  miR-29 is required for normal endothelial function in human and rat arterioles.**

A   Intraluminal transfection of anti-miR-29b-3p substantially impaired acetylcholine (Ach)-induced vasodilation in human arterioles. $N = 5$, *$P < 0.05$, two-way ANOVA; #$P < 0.05$ at the given concentration of Ach, Holm–Sidak test.

B   Intraluminal transfection of anti-miR-29a-3p or anti-miR-29b-3p impaired Ach-induced vasodilation in rat arterioles. $N = 4$ for scrambled anti-miR, 5 for anti-miR-29a-3p, and 5 for anti-miR-29b-3p. *$P < 0.05$, two-way ANOVA; #$P < 0.05$ at the given concentration of Ach compared with scrambled anti-miR, Holm–Sidak test.

C   Generation of *Mir29b-1/a$^{-/-}$* rats using a Transcriptional Activator-Like Effector Nucleases (TALEN) method. Four nucleotides overlapping with the seed region of miR-29b-3p were deleted in the SS-Chr 13BN-*Mir29b1*$^{em1Mcwi}$ (i.e., *Mir29b-1/a* mutant or *Mir29b-1/a$^{-/-}$*) rat.

D   Extraction of an endothelium-enriched fraction (EC elute) from rat gluteal arterioles. PECAM-1 and MYH11 are marker genes for endothelial and smooth muscle cells, respectively. $n = 17$, *$P < 0.05$ vs. the remainder of the vessel (paired $t$-test).

E   Abundance of miR-29b-3p and miR-29a-3p in the EC elute from the gluteal arterioles of *Mir29b-1/a$^{-/-}$* rats. $n = 5$ for miR-29a and 6 for miR-29b, *$P < 0.05$ vs. *Mir29b-1/a$^{+/+}$*, #$P < 0.05$ vs. *Mir29b-1/a$^{+/-}$*, one-way ANOVA followed by Holm–Sidak test.

F   Ach-induced dilation of gluteal resistance arterioles was substantially attenuated in *Mir29b-1/a* mutant rats. $N = 9$ for *Mir29b-1/a$^{+/+}$*, 10 for *Mir29b-1/a$^{+/-}$*, and 7 for *Mir29b-1/a$^{-/-}$*; *$P < 0.05$, two-way ANOVA; #$P < 0.05$ vs. *Mir29b-1/a$^{+/+}$* at the given concentration of Ach, &$P < 0.05$ vs. *Mir29b-1/a$^{+/+}$* and *Mir29b-1/a$^{+/-}$* at the given concentration of Ach, Holm–Sidak test.

G   The development of hypertension was significantly exacerbated in *Mir29b-1/a$^{-/-}$* rats. Rats were approximately 7–8 weeks old on day 1. $N = 10$ for *Mir29b-1/a$^{+/+}$*, 11 for *Mir29b-1/a$^{+/-}$*, and 6 for *Mir29b-1/a$^{-/-}$*; *$P < 0.05$ vs. *Mir29b-1/a$^{+/+}$*, two-way ANOVA followed by Holm–Sidak test.

Data information: Data are reported as mean ± SEM.

We developed *Mir29b-1/a* mutant rats to further examine the role of miR-29 in normal endothelial function. miR-29b is encoded by *Mir29b-1* and *Mir29b-2* genes. The *Mir29b-1* gene is located in close genomic proximity to *Mir29a* gene. We used a Transcriptional Activator-Like Effector Nucleases (TALEN) method to target the *Mir29b-1/a* gene on the genetic background of SS-Chr13$^{BN}$ rats (Geurts *et al*, 2010). SS-Chr13$^{BN}$ rats are inbred rats that have been used as normal control in studies of vascular dysfunction (Cowley *et al*, 2001; Drenjancevic-Peric *et al*, 2003). We identified and established a colony of rats with deletion of four base pairs in the genomic segment of the *Mir29b-1/a* gene that encodes nucleotides 6–9 in the sequence of mature miR-29b-3p (Fig 2C). The four nucleotides deleted overlapped with the seed region (nucleotides 2–7) that is critical for the canonical function of miR-29b-3p.

Gluteal arterioles isolated from the rats were perfused with a lysing solution, and the eluted fraction and the remainder of the vessel were collected. The eluted fraction was enriched for PECAM-1, an endothelial marker gene, and largely depleted of MYH11, a smooth muscle marker gene, indicating the eluted fraction was enriched for endothelial lysate (Fig 2D). miR-29b-3p abundance in the endothelium-enriched fraction of the gluteal arterioles was significantly reduced in heterozygous mutant rats and further reduced to approximately 8% of the wild-type level in homozygous mutant rats (Fig 2E). Similar expression patterns were observed in the endothelium-enriched fraction obtained from the aorta (Appendix Fig S4). The residual miR-29b-3p might be expressed from the separate *Mir29b-2* gene. The disruption of the miR-29b-3p strand in the mutant rat could change the level of the passenger strand miR-29b1-5p by influencing the formation or stability of the miR-29b1 hairpin structure. However, miR-29b1-5p was not detectable in the endothelium-enriched fractions even in the wild-type rats and remained undetectable in the mutant rats. Levels of miR-29a-3p, which is co-transcribed with miR-29b1, were also decreased in the mutant rat (Fig 2E), suggesting the four base pair deletion likely destabilized the primary transcript that contains the hairpins of both miR-29a and miR-29b1. The passenger strand miR-29a-5p was not detectable in any of the samples. Levels of miR-29c-3p, which is transcribed from a separate gene, were not significantly different between mutant rats and wild-type littermates. The abundance of miR-29 isoforms in the vessel remainder, as well as their pattern of changes in the mutant rats, was similar to the endothelium-enriched fraction. Taken

together, these data indicated the mutant rat, which we designated *Mir29b-1/a$^{-/-}$*, was a model of robust miR-29a-3p and miR-29b-3p knockdown.

Acetylcholine-induced vasodilation in cutaneous branches of the inferior gluteal artery was substantially impaired in *Mir29b-1/a$^{-/-}$* rats and, to a lesser extent, *Mir29b-1/a$^{+/-}$* rats compared to wild-type littermates (Fig 2F). Peak Ach dilation was reduced from $83 \pm 4\%$ in wild-type littermates to $60 \pm 6\%$ in *Mir29b-1/a$^{+/-}$* rats and $40 \pm 3\%$ in *Mir29b-1/a$^{-/-}$* rats. There were no differences in papaverine response ($96 \pm 2\%$, $91 \pm 3\%$, and $95 \pm 3\%$, respectively). In addition, the NO donor spermine NONOate induced robust dilation in arterioles from *Mir29b-1/a$^{-/-}$* rats (Appendix Fig S5). The effect of the *Mir29b-1/a* mutation on rat arterioles mirrors the effect of anti-miR-29 on human and rat arterioles (Fig 2A and B) and supports the conclusion that miR-29 is required for normal vascular endothelial function as measured by acetylcholine-induced vasodilation.

Endothelial dysfunction could lead to elevation of arterial blood pressure. Mean arterial blood pressure, measured via indwelling catheters in conscious, freely moving rats, was indeed significantly higher in *Mir29b-1/a$^{-/-}$* rats compared to wild-type littermates (Fig 2G).

## miR-29 contributes to endothelial function by increasing NO

Nitric oxide plays a dominant role in mediating acetylcholine-induced vasodilation in human gluteal arterioles, which were the vessels used in this study (Dharmashankar *et al*, 2012). Vasodilation of these vessels to acetylcholine was not impaired by inhibition of cyclooxygenase with indomethacin, hydrogen peroxide with PEG-catalase, or epoxyeicosatrienoic acids with 14,15-epoxyeicosa-5(Z)-enoic acid.

We examined whether the beneficial effect of miR-29 on endothelium-dependent vasodilation was mediated by increasing NO. Intraluminal transfection with miR-29b-3p mimic significantly increased NO levels, measured as DAF-2 fluorescence intensity, in gluteal arterioles from T2DM patients (Fig 3A). The miR-29b-induced increase in NO was blocked by L-NAME, an inhibitor of endothelium-derived NO synthase (eNOS), suggesting the miR-29b induced in NO levels in these vessels was nitric oxide synthase-dependent. Intraluminal treatment with anti-miR-29b-3p significantly reduced NO levels in gluteal arterioles from non-DM human subjects (Fig 3B). Similarly,

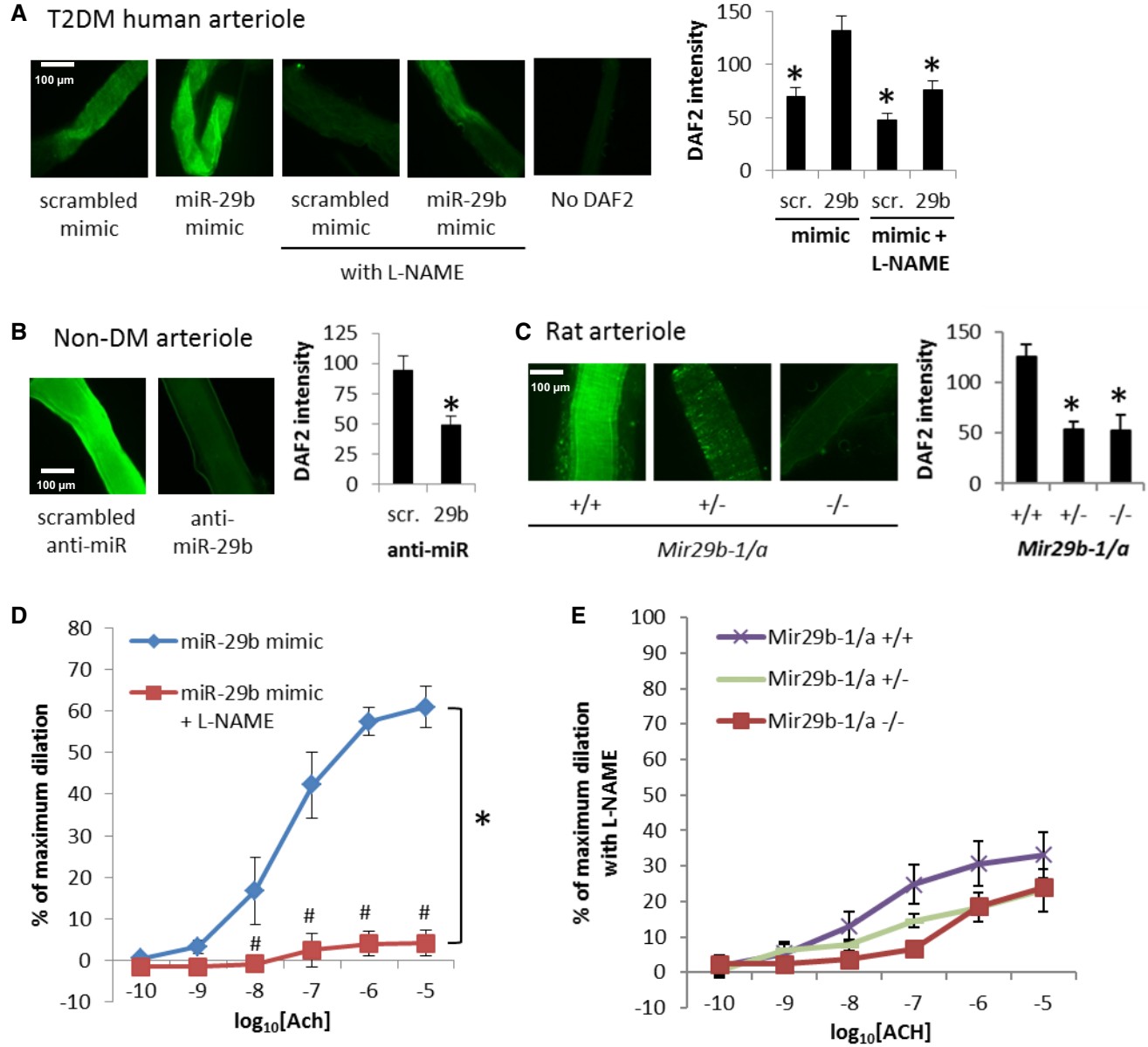

**Figure 3. miR-29 contributes to endothelial function by increasing NO levels.**

A   Intraluminal transfection with miR-29b-3p mimic increased NO levels, measured as DAF2 intensity, in arterioles from T2DM patients, an effect attenuated by eNOS inhibitor L-NAME. $N = 5$, $P < 0.001$ overall by RM ANOVA, *$P < 0.001$ for vs. miR-29b-3p mimic (Holm-Sidak test).

B   Intraluminal transfection with anti-miR-29b reduces NO levels in normal human gluteal arterioles. $N = 6$, *$P < 0.05$, signed rank test.

C   NO levels in gluteal arteriole were reduced in *Mir29b-1/a* mutant rats. $N = 11$ for *Mir29b-1/a$^{+/+}$*, 9 for *Mir29b-1/a$^{+/-}$*, and 6 for *Mir29b-1/a$^{-/-}$*; *$P < 0.05$ vs. *Mir29b-1/a$^{+/+}$*, two-way ANOVA followed by Holm–Sidak test.

D   L-NAME blocked acetylcholine (Ach)-induced vasodilation in arterioles from T2DM patients that had been restored by intraluminal transfection with miR-29b-3p mimic. $N = 4$, *$P < 0.05$, two-way ANOVA; #$P < 0.05$ at the given concentration of Ach, Holm–Sidak test.

E   Ach-induced vasodilation in gluteal arterioles from *Mir29b-1/a$^{-/-}$* rats in the presence of L-NAME. Data without L-NAME, which were obtained in parallel with experiments with L-NAME and under otherwise similar conditions, were shown on the same scale in Fig 2F. $N = 8$ for *Mir29b-1/a$^{+/+}$*, 7 for *Mir29b-1/a$^{+/-}$*, and 7 for *Mir29b-1/a$^{-/-}$*.

Data information: Data are reported as mean ± SEM.

NO levels were significantly reduced in gluteal arterioles obtained from *Mir29b-1/a$^{+/-}$* and *Mir29b-1/a$^{-/-}$* rats compared to wild-type littermates (Fig 3C). These data indicate miR-29 contributes to NO levels in non-DM arterioles and can restore normal NO levels in T2DM arterioles.

The restoration of acetylcholine-induced vasodilation in gluteal arterioles from T2DM patients by miR-29b-3p mimic was abolished by L-NAME (Fig 3D). L-NAME also substantially attenuated acetylcholine-induced vasodilation in gluteal arterioles from wild-type rats (Fig 3E, compared to the data without L-NAME shown in Fig 2F). In

the presence of L-NAME, the differences in acetylcholine-induced vasodilation between $Mir29b\text{-}1/a^{+/+}$, $Mir29b\text{-}1/a^{+/-}$, and $Mir29b\text{-}1/a^{-/-}$ rats (shown in Fig 2F) were largely abolished (Fig 3E). These data, together with the DAF-2 DA data shown above, indicate the beneficial effect of miR-29 on endothelial function is mediated by increasing NO.

## miR-29 regulates genes related to NO levels

We examined potential molecular mechanisms by which miR-29 regulates NO levels. NO production in vascular endothelial cells is mediated by endothelial nitric oxide synthase (eNOS). eNOS is not a predicted target gene of miR-29. Protein abundance of eNOS in mesentery arteries was not significantly different between $Mir29b\text{-}1/a^{-/-}$ rats and wild-type littermates (Appendix Fig S6).

Nitric oxide production can be altered by changes in eNOS activity. Moreover, NO levels are determined by the balance between NO production and destruction. Scavenging by reactive oxygen species is a major mechanism for NO destruction. Analysis of Gene Ontology and pathway databases yielded 286 genes that were known to be involved in the regulation of NO or reactive oxygen species levels (Dataset EV2; Gene Ontology, 2015). Twelve of the 286 genes are predicted target genes of miR-29 (Dataset EV2).

We performed RNA-seq analysis in gluteal arterioles isolated from $Mir29b\text{-}1/a^{-/-}$ rats ($n = 5$) and wild-type littermates ($n = 6$; Dataset EV3). The expression profiles of 18,615 detected genes correctly clustered the samples by genotype, indicating the $Mir29b\text{-}1/a$ mutation had substantial and reproducible effects on the gene expression profile in gluteal arterioles (Appendix Fig S7). Comparison of $Mir29b\text{-}1/a^{-/-}$ and wild-type littermates identified 4,094 genes with $P < 0.05$, 1,852 of which had $P < 0.05$ following adjustment for multiple comparisons. The differentially expressed genes were enriched for genes involved in several pathways and functional processes, including vascular structure and function, accordingly to a Gene Ontology enrichment analysis (Dataset EV4).

Of the 286 genes relevant to NO regulation (Dataset EV2), 179 were detected in the RNA-seq analysis, of which 32 were differentially expressed between $Mir29b\text{-}1/a^{-/-}$ and wild-type littermates (adjusted $P < 0.05$; Fig 4A). The proportion (32 of 179, or 18%) was nearly two times higher than when all detected genes were considered (1,852 of 18,615, or 10%; Fig 4B). The enrichment of NO-related genes in the differentially expressed genes was statistically extremely significant (chi-squared test, $P < 10^{-12}$).

Of the 12 predicted miR-29 target genes with known involvement in NO regulation (Dataset EV2), lysophospholipase I (Lypla1) was up-regulated from 65 FPKM in wild-type rats to 104 FPKM in $Mir29b\text{-}1/a^{-/-}$ rats ($P = 0.005$, adjusted $P = 0.05$; Fig 4A). Lypla1 can reduce eNOS activity via depalmitoylation of eNOS (Yeh et al, 1999). Treatment of cultured human dermal microvascular endothelial cells (hMVEC-d) with miR-29b-3p mimic resulted in significant down-regulation of Lypla1 protein (Fig 4C). Lypla1 3′-UTR contains a highly conserved binding site for miR-29-3p (Fig 4D). miR-29b-3p mimics decreased luciferase reporter activities when the luciferase sequence was followed by the 3′-UTR of either human or rat Lypla1 mRNA (Fig 4E). Mutations of 1 or 2 nucleotides or deletion of 4–6 nucleotides in the binding site in the 3′-UTRs abrogated the effect of miR-29b-3p on reporter activities (Fig 4D and E). These data indicated Lypla1 was a direct target of miR-29b-3p.

Neuromodulin (Gap43), another predicted target of miR-29 that can reduce eNOS activation by inhibiting calcium/calmodulin, was up-regulated from 2.9 FPKM in wild-type rats to 6.8 FPKM in $Mir29b\text{-}1/a^{-/-}$ rats ($P = 0.04$) according to the RNA-seq analysis of gluteal arterioles (Dataset EV3). Real-time PCR analysis did not reproducibly detect Gap43 mRNA in the small amount of gluteal arteriole samples but confirmed Gap43 mRNA was up-regulated in the carotid artery of $Mir29b\text{-}1/a^{-/-}$ rats (Appendix Fig S8).

## Targeting of Lypla1 contributes to the effect of miR-29 on NO and endothelium-dependent vasodilation

We examined the functional role of Lypla1 targeting in the effect of miR-29 on NO and endothelium-dependent vasodilation, given the reported role of Lypla1 in the depalmitoylation of eNOS, which would reduce eNOS activity (Yeh et al, 1999). We first examined whether targeting of Lypla1 contributed to the effect of miR-29 in enhancing eNOS activity. eNOS activity was measured as calcium ionophore A23187-induced, L-NAME-inhibitable production of NO based on the intensity of a fluorescent indicator loaded into the cells or levels of NO metabolites nitrite and nitrate in the culture supernatant. Knockdown of Lypla1 with an siRNA (Fig 5A) resulted in increases in A23187-induced, L-NAME-inhibitable production of NO in hMVEC-d cells (Fig 5B), indicating Lypla1 indeed inhibited eNOS activities in these cells. Similarly, treatment with miR-29b-3p mimics, which would down-regulate Lypla1 (see Fig 4C), resulted in significant increases in eNOS activity (Fig 5C). Furthermore, the effect of miR-29b-3p mimics in promoting eNOS activities was abrogated by pre-transfection of the cells with a Lypla1 expression construct but not the control expression vector (Fig 5D). The 3′-UTR of Lypla1 was omitted from the Lypla1 over-expression construct to avoid targeting by miR-29. The Lypla1 expression construct did not significantly decreased eNOS activity in cells transfected with the scrambled mimic. Conversely, treatment of the cells with anti-miR-29b resulted in up-regulation of Lypla1 mRNA and decreases in eNOS activity (Fig 5E and F). The decrease in eNOS activity following anti-miR-29b treatment was attenuated by simultaneous treatment with Lypla1 siRNA (Fig 5F). These data indicate targeting of Lypla1 contributes to the effect of miR-29 in enhancing eNOS activity and promoting NO production.

We next examined the role of Lypla1 in the effect of miR-29 in promoting endothelium-dependent vasodilation. Gluteal arterioles obtained from $Mir29b\text{-}1/a^{-/-}$ rats were transfected intraluminally with an Lypla1 siRNA or a scrambled control siRNA. Arterioles transfected with scrambled siRNA showed poor Ach-induced dilation (Fig 5G), similar to what was shown in Fig 2F. Lypla1 siRNA significantly improved Ach-induced dilation in $Mir29b\text{-}1/a^{-/-}$ arterioles (Fig 5G). In gluteal arterioles obtained from T2DM patients, in which miR-29 function was compromised, Lypla1 mRNA was up-regulated (Fig 5H). Intraluminal transfection of Lypla1 siRNA significantly improved Ach-induced dilation in T2DM patient arterioles (Fig 5I). Arterioles obtained from non-DM subjects and treated with anti-miR-29b showed impaired Ach-induced dilation (Fig 5J), similar to what was shown in Fig 2A. Simultaneous intraluminal transfection with Lypla1 siRNA significantly improved Ach-induced dilation in these arterioles (Fig 5J). In all treatment groups shown in Fig 5I and J, treatment with L-NAME largely abolished Ach-induced dilation. These data indicate targeting of Lypla1 contributes to the

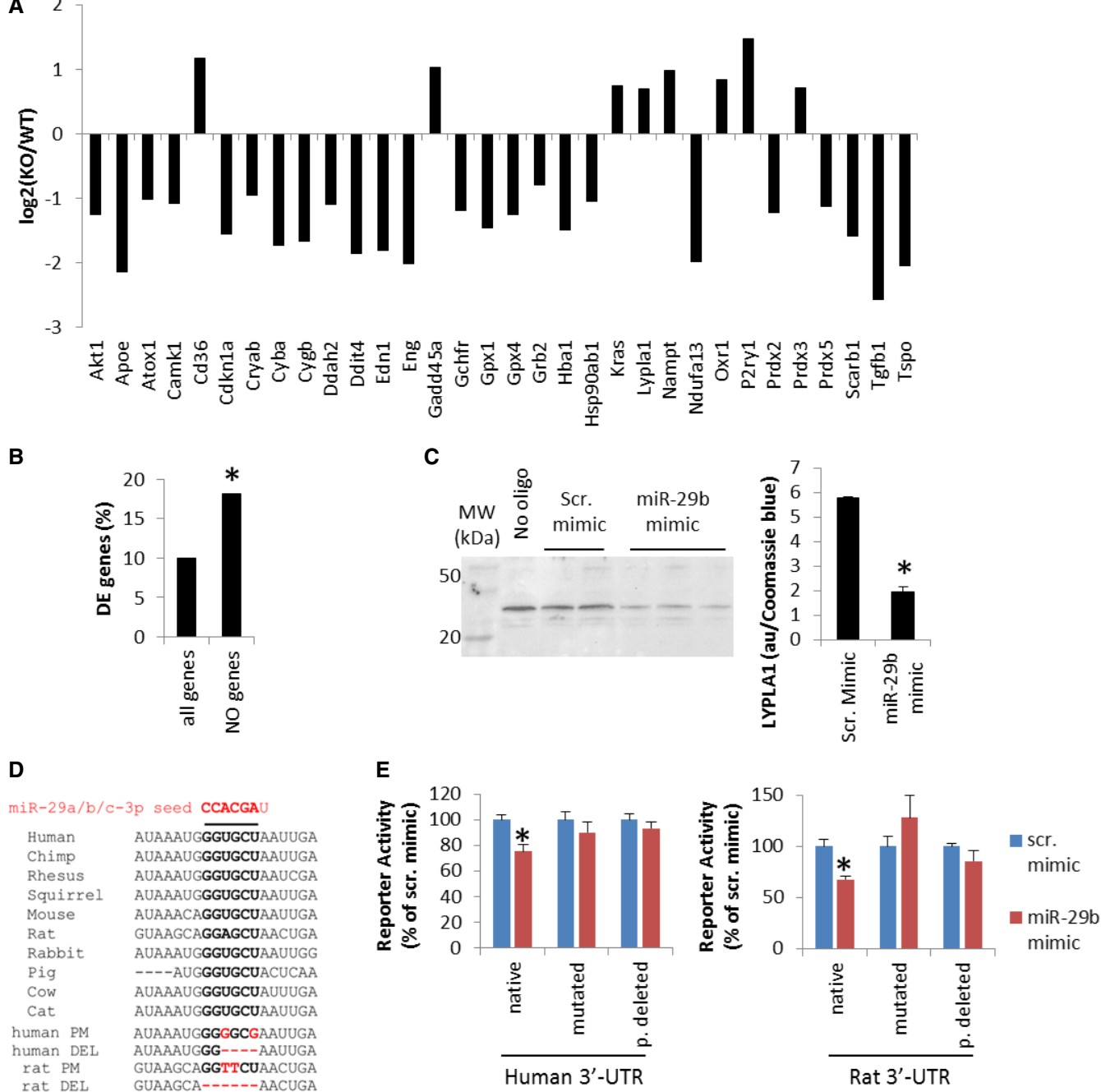

**Figure 4.  miR-29 regulates genes involved in determining NO levels, including Lypla1.**

A    *Mir29b-1/a* mutation in rats led to differential expression of several genes relevant to the regulation of NO bioavailability in gluteal arterioles. Data were from RNA-seq analysis. The genes shown were differentially expressed between *Mir29b-1/a*$^{-/-}$ rats (KO) and wild-type (WT) littermates with adjusted *P*-values < 0.05.

B    *Mir29b-1/a* mutation preferentially influenced genes relevant to the regulation of NO bioavailability. See the text for an explanation of how the percentage values were calculated. *$P < 10^{-12}$, chi-squared test.

C    miR-29b-3p mimic down-regulated Lypla1 in human dermal microvascular endothelial (HMVEC-d) cells. Lypla1 abundance was normalized by Coomassie blue staining of the membrane. *N* = 2 for scrambled mimic and 3 for miR-29b mimic, *$P < 0.05$, unpaired *t*-test.

D    Conserved binding site for miR-29-3p seed region in the Lypla1 3′-untranslated region (3′-UTR) and the mutated (PM) or partially deleted (DEL) UTR constructs.

E    miR-29b-3p mimic decreased luciferase activity under the control of a human or rat Lypla1 3′-UTR segment. The effect was abrogated with mutated or partially deleted (p. deleted) UTRs. *N* = 5 for human native and rat mutated or partially deleted, *n* = 10 for other groups, *$P < 0.05$, unpaired *t*-test. Data are reported as mean ± SEM.

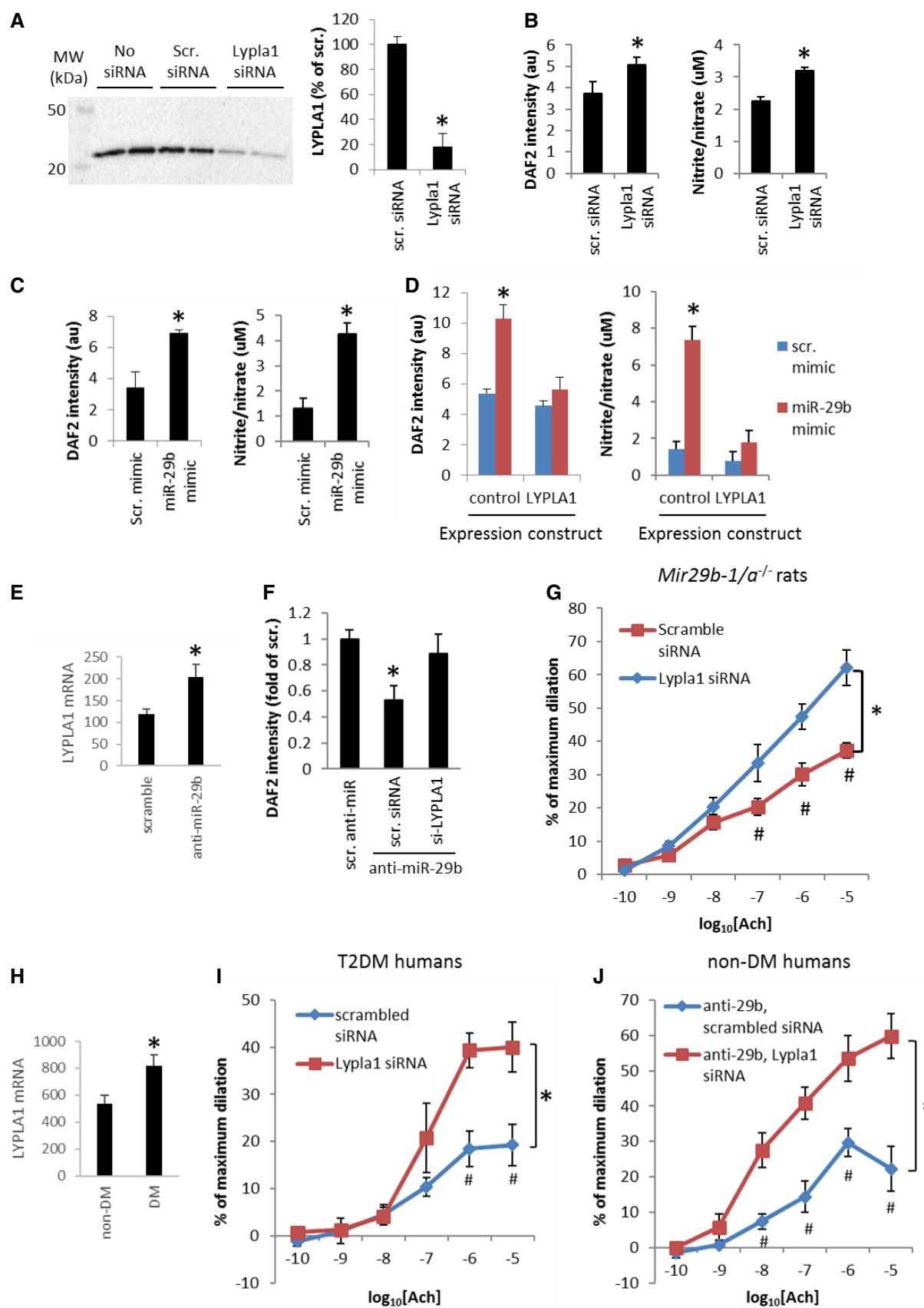

**Figure 5.**

◀

**Figure 5.  Targeting of Lypla1 contributes to the effect of miR-29 in promoting NO generation and endothelium-dependent vasodilation.**

A   Lypla1 siRNA knocked down Lypla1 in HMVEC-d cells. Lypla1 abundance was normalized by Coomassie blue staining of the membrane or β-actin. $N = 4$, *$P < 0.05$, unpaired *t*-test.

B   Lypla1 siRNA increased calcium ionophore A23187-induced, L-NAME-inhibitable NO production in HMVEC-d cells. NO levels were measured as DAF2 fluorescent signal (arbitrary unit) in the cells or nitrite and nitrate concentrations (μM) in culture supernatant. $N = 24$ for DAF2 and $n = 16$ for nitrite/nitrate, *$P < 0.05$, unpaired *t*-test.

C   miR-29b-3p mimic increased calcium ionophore A23187-induced, L-NAME-inhibitable NO production in HMVEC-d cells. $N = 8$ for DAF2 and $n = 6$ for nitrite/nitrate, *$P < 0.05$, unpaired *t*-test.

D   Lypla1 over-expression abrogated the effect of miR-29b-3p on NO in HMVEC-d cells. The cells were transfected with plasmids containing Lypla1 open reading frame or a control plasmid before treatment with miRNA mimics. $N = 12$, *$P < 0.05$ vs. scrambled mimic or miR-29b-3p mimic in the presence of Lypla1 (two-way ANOVA followed by Holm-Sidak test).

E   Anti-miR-29b-3p treatment in HMVEC-d cells resulted in up-regulation of Lypla1 mRNA. $N = 12$, *$P < 0.05$, unpaired *t*-test.

F   Lypla1 siRNA (si-Lypla1) attenuated the effect of anti-miR-29b-3p on NO in HMVEC-d cells. $N = 9$, *$P < 0.05$ vs. scrambled anti-miR (one-way ANOVA followed by Holm-Sidak test).

G   Intraluminal transfection of Lypla1 siRNA improved acetylcholine (Ach)-induced vasodilation in arterioles from *Mir29b-1/a*$^{-/-}$ rats. $N = 5$ scrambled siRNA and 6 Lypla1 siRNA, *$P < 0.05$, two-way ANOVA; #$P < 0.05$ at the given concentration of Ach, Holm–Sidak test.

H   Lypla1 mRNA was up-regulated in arterioles from DM patients. $N = 17$ non-DM and 13 DM, *$P < 0.05$, unpaired *t*-test.

I   Intraluminal transfection of Lypla1 siRNA improved Ach-induced vasodilation in arterioles from T2DM patients. $N = 5$, *$P < 0.05$, two-way ANOVA; #$P < 0.05$ at the given concentration of Ach, Holm–Sidak test.

J   Intraluminal transfection of Lypla1 siRNA improved Ach-induced vasodilation in arterioles from non-DM humans and treated with anti-miR-29b-3p. $N = 5$, *$P < 0.05$, two-way ANOVA; #$P < 0.05$ at the given concentration of Ach, Holm–Sidak test.

Data information: Data are reported as mean ± SEM.

effect of miR-29 on endothelium-dependent vasodilation in both human and rat arterioles.

## Discussion

Using human tissues with clinical relevance and challenging to obtain, as well as the first miRNA gene mutant rat model that we are aware of, we have (i) established miRNA expression patterns in human resistance arterioles associated with endothelial dysfunction in T2DM, (ii) discovered an important role of miR-29 in promoting NO production and endothelial function, and (iii) identified miR-29 as a potent agent for restoring endothelium-dependent vasodilation in human T2DM arterioles through a mechanism that in significant part involves modulation of Lypla1 expression.

Several prior studies report miRNA profiles of patients with DM or metabolic syndrome compared to non-DM, non-insulin-resistant subjects. These have been performed using qPCR analysis of either plasma or mononuclear cells (Zampetaki *et al*, 2010; Kong *et al*, 2011; Karolina *et al*, 2012; Ortega *et al*, 2014). Our miRNA expression data significantly extend these data by (i) measuring miRNA expression in the structure (arteriole) directly responsible for key complications in diabetes, and (ii) using next-generation sequencing to quantify whole genome miRNA expression. Cross-referencing these prior works with the top 30 miRNAs found in the current study identifies seven miRNAs that appear differentially expressed in both plasma and small arteries: miR-21, miR-29a, miR-29b, miR-99b, miR 126, miR 140-5p, and 195-3p. These data suggest this subset of miRNAs might represent, at least in part, the vascular-specific miRNA contribution or "signature" in human plasma. All of these miRNAs have been implicated in vascular disease (Ito *et al*, 2010; Raitoharju *et al*, 2011; Kane *et al*, 2012; Kriegel *et al*, 2012; Wang *et al*, 2012b; Jansen *et al*, 2013; Zhu *et al*, 2013; Mortuza *et al*, 2014). These miRNAs regulate mRNA for proteins involved in angiogenesis; endothelial cell differentiation, repair, and senescence; vascular smooth muscle proliferation and migration; and vascular extracellular matrix structure.

While all seven aforementioned miRNAs are expressed at higher levels in DM arterioles, only two of these miRNAs (miR-29a,

miR-140-5p) are concordantly expressed at higher levels in DM plasma. Four other miRNAs (miR-21, miR-29b, miR-126, and miR-195) are expressed at lower levels in DM plasma. (miR-99b is known to be differentially expressed in plasma but its directionality is not reported.; Zampetaki *et al*, 2010; Kong *et al*, 2011; Karolina *et al*, 2012; Ortega *et al*, 2014). miR-146b, while known to be over-expressed in atherosclerotic plaques, has not been demonstrated to be differentially expressed in plasma (Raitoharju *et al*, 2011). Moreover, 23 of the top 30 miRs found in human arterioles have not been reflected in plasma studies (Fang *et al*, 2010; Boon *et al*, 2012; Zitman-Gal *et al*, 2014). Some of the 23 miRNAs have known vascular regulatory effects (Fang *et al*, 2010; Raitoharju *et al*, 2011; Kriegel *et al*, 2012; Kumarswamy *et al*, 2014). The reasons behind these differences in vascular expression pattern relative to plasma expression are unclear and likely multifactorial. Possible physiological explanations include (i) miRNA-specific differences in packaging and release into the circulation, (ii) differences in relative uptake of miRNAs systemically, (iii) miRNA-specific differences in disposal based on systemic and/or local factors, and (iv) non-vascular sources of plasma miRNAs. These findings suggest the presence of pathways that may have profound significance regarding the auto-crine, paracrine, and endocrine actions of vascular miRNAs and merit further investigation. The findings also highlight the critical value of understanding miRNA expression profiles in tissues directly relevant to vascular pathology, such as small arteries.

No miRNA has been shown to have a potent effect in restoring endothelium-dependent vasodilation in disease prior to the present study. A few miRNAs, such as miR-155 and miR-204, have been shown to contribute to the impairment of endothelium-dependent vasodilation induced by gut microbiome and inflammatory cytokines (Sun *et al*, 2012; Vikram *et al*, 2016) Additional miRNAs including miR-126-5p and miR-132 might be involved in other aspects of endothelial function such as endothelial cell proliferation and angiogenesis (Anand *et al*, 2010; Schober *et al*, 2014). The miR-29 family of mature miRNAs includes miR-29a, miR-29b, and miR-29c. The three isoforms are encoded by two clusters of gene, one including *Mir29a* and *Mir29b-1* and the other including *Mir29b-2* and *Mir29c*. The best understood functions of miR-29 include antifibrotic effects in several organs and effects on promoting apoptosis and regulating

cell differentiation (Kriegel *et al*, 2012). miR-29 targeting of approximately 20 extracellular matrix genes in several organs is one of the most dramatic examples of a miRNA directly targeting a large number of genes that belong to a functional pathway (Liu *et al*, 2010). The role of miR-29 in endothelial function was largely unknown prior to the present study.

Homeostatic levels of NO are central to maintaining normal vascular tone, inhibit inflammation and cellular proliferation, and balance fibrinolytic and thrombotic factors (Widlansky *et al*, 2003). Endothelial nitric oxide synthase (eNOS) is the primary source of endothelial NO. Inducible NOS could be elevated in DM vessels given the inflammatory condition. However, the Ach-stimulated vasodilation examined in the present study should primarily involve eNOS since Ach activates eNOS via a calcium-dependent process and that iNOS activity is calcium-independent in the endothelium. NO production by eNOS is highly regulated, depending on enzyme expression levels, phosphorylation states, glycosylation, calcium levels, oxidation state of co-factors, and, importantly, intracellular location (Dudzinski *et al*, 2006). Production of NO by eNOS requires eNOS to be anchored to the plasma membrane in caveolae (Shaul *et al*, 1996). Localization of eNOS to the caveolae in the plasma membrane requires post-translational N-terminal myristoylation, and subsequent palmitoylation at N-terminal cysteine residues 15 and 26 further strengthens eNOS's bond to the plasma membrane caveolae (Gonzalez *et al*, 2002; Shaul, 2002). Palmitoylation of eNOS is dynamically regulated, impacting the ability of the endothelial cell to produce NO via eNOS (Yeh *et al*, 1999). Lypla1 depalmitoylates eNOS, which can reduce eNOS-dependent NO production (Yeh *et al*, 1999). The targeting of Lypla1 by miR-29 discovered in the present study provides one of the possible mechanisms by which miR-29 can promote NO production, a mechanism we demonstrate can be blocked by Lypla1 over-expression. It is equally interesting that, at the transcriptome level, miR-29 appears to preferentially regulate a large number of genes that can potentially alter NO bioavailability. Most of these genes, however, are not direct targets of miR-29. It remains to be determined whether miR-29 indeed coordinately regulates these genes to influence NO levels and what the mechanisms that enable such coordinated regulation to occur are.

miR-29 isoforms have identical seed region sequence and are predicted to target nearly identical sets of genes (Kriegel *et al*, 2012). Mimics of both miR-29a-3p and miR-29b-3p significantly improved endothelial function in human T2DM arterioles. It suggests improvement in endothelial function is likely a canonical effect of the miR-29 family. It is noteworthy that anti-miR-29b oligonucleotide substantially impaired endothelial function in spite of the fact that miR-29b was less abundant than miR-29a in the arterioles analyzed. It is possible that non-canonical actions of miR-29b not shared by miR-29a amplify the effect of miR-29b on endothelial function (Hwang *et al*, 2007), which would be valuable to examine in the future studies. It is also possible that anti-miR-29b inhibited both miR-29b and miR-29a, which have highly similar sequences. An anti-miR could induce degradation of target miRNA, but could also just block target miRNA, making it difficult to test the specificity of anti-miR by measuring the abundance of target miRNA. What is clear, however, is that anti-miR-29b was capable of influencing NO in arterioles and cultured cells, which is a mechanism that involves canonical miR-29 target gene Lypla1 as indicated by the cell culture experiment. Four nucleotides overlapping with

the seed region were deleted in the *Mir29b-1/a* mutant rat we developed. While unlikely, the possibility that a mutant form of miR-29b-3p is produced and influences endothelial function through pathways independent of miR-29 cannot be ruled out. The regulatory system is, as expected, complex, and some of our experiments separately could have alternative interpretations. Taken together, however, our data from human vessels, mutant rats, and cultured cells reinforce each other and support the conclusion that miR-29 contributes to normal endothelial function and can restore endothelium-dependent vasodilation in disease.

We described the disease subjects in the present study as T2DM for simplicity. T2DM often presents as part of a metabolic syndrome that includes hypertension and dyslipidemia. This was evident in the group of subjects we studied as shown in Appendix Tables S1 and S3. A multivariate analysis to identify the specific contribution of T2DM to the differential expression of miRNAs is not practical given the sample size. However, such a constellation of cardiometabolic disorders is common in patients, which supports the broad clinical relevance of our findings. The mechanism by which T2DM or its associated diseases lead to the differential expression of miRNAs in arterioles remains to be examined in the future studies. The focus of the present study was on the functional and disease relevance of miR-29 in arterioles.

The potent effect of miR-29 mimics in restoring endothelium-dependent vasodilation in T2DM arterioles supports the potential value of developing miR-29 mimics as a therapeutic for microvascular complications of T2DM as well as many other cardiovascular diseases where endothelial dysfunction plays a critical pathophysiological role. The well-established effect of miR-29 in reducing extracellular matrix accumulation could add additional benefits to a miR-29 therapy since fibrosis is a common pathological change in cardiovascular disease (van Rooij *et al*, 2008; Liu *et al*, 2010; Montgomery *et al*, 2014). It is, however, important to recognize that the extracellular matrix is physiologically important and excessive reduction in extracellular matrix by miR-29 could compromise normal tissue structure or repair mechanisms, leading to aneurysm formation or destabilization of atherosclerotic plaques (Boon *et al*, 2011; Maegdefessel *et al*, 2012; Ulrich *et al*, 2016). miR-29 might also contribute to the development of pulmonary arterial hypertension via its effects on energy metabolism (Chen *et al*, 2016). It would be important to examine possible side effects of a miR-29 therapy as well as challenges associated with the delivery and targeting of miRNA mimics (Montgomery *et al*, 2014).

## Materials and Methods

### Human subjects

We recruited a total of 108 subjects (52 with T2DM, 56 non-diabetic controls) aged 35–70 years for this study. Of these subjects, 38 (18 with T2DM, 20 controls) were involved in the initial characterization and small RNA deep sequencing analysis, and the remaining 70 subjects as well as five of the initial subjects were involved in the studies of miR-29. The study protocol was conducted in accordance with a protocol approved by the Medical College of Wisconsin's Institutional Review Board. Informed consent was obtained from all subjects and that the experiments conformed to the principles set

out in the WMA Declaration of Helsinki and the Department of Health and Human Services Belmont Report. The diagnosis of T2DM was confirmed by clinical history. Non-DM controls underwent a screening visit to confirm the absence of prevalent macrovascular disease (no history of myocardial infarction, stroke, or peripheral vascular disease by medical history), hypertension (resting blood pressure $\geq$ 140/90 mmHg based on an average of measurements in triplicate or taking antihypertensive medications), hyperlipidemia (LDL cholesterol $\geq$ 160 mg/dl or taking lipid-lowering medications), and diabetes based on current diagnostic criteria. All potential subjects with a history of active smoking within 1 year of enrollment were excluded.

### General procedures for the human study

All subjects fasted for at least 6 h prior to any study procedures. Following an intake medical history, height and weight were measured in centimeters and kilograms, respectively. Standing waist circumference was measured at the level of the umbilicus. Heart rate and blood pressure (BP) were measured in triplicate and averaged. Venous blood samples were drawn from a peripheral arm or forearm vein.

### Brachial artery reactivity testing, gluteal adipose pad biopsy, harvesting of resistance arterioles, and measurement of *in vitro* endothelial function by videomicroscopy for non-transfected arterioles

*In vivo* measurements of endothelial function were performed by flow-mediated dilation (FMD) as previously described (Kizhake-kuttu *et al*, 2010, 2012; Babar *et al*, 2011; Wang *et al*, 2012a). All subjects underwent gluteal adipose pad biopsy to obtain adipose arterioles for miRNA expression analyses and functional studies as previously described (Dharmashankar *et al*, 2012; Kizhakekuttu *et al*, 2012; Wang *et al*, 2012a; Suboc *et al*, 2013). Greater detail is included in the Appendix.

### Intraluminal transfection of isolated human resistance arterioles and measurement of vasodilation

Isolated arterioles were transferred to a culture myograph chamber (DMT, 204CM) and cannulated with tapered glass micropipettes filled with the Krebs buffer consisting of (in mM) 123 NaCl, 4.7 KCl, 2.5 $CaCl_2$, 0.026 EDTA, 1.2 $MgSO_4$, 20 $NaHCO_3$, 1.2 $KH_2PO_4$, and 5 glucose. After cannulating one end of the arteriole with a glass micropipette, anti-miR-29b-3p, miR-29a-3p mimic, miR-29b-3p mimic, or their respective control scrambled oligonucleotide was injected into the lumen of the arteriole. Locked nucleic acid (LNA)-modified anti-miR oligonucleotides were from Exiqon and used at the final concentration of 133 nM in the Krebs buffer. miRNA mimic oligonucleotides were from Thermo Fisher and prepared for Lipofectamine 2000-mediated transfection at the final concentration of 20 nM following instructions from the manufacturer. The other end of the arteriole was mounted on a glass micropipette, and the chamber was transferred to the stage of a DMT culture myograph system (204CM, DMT, Denmark) attached to a video camera and imaged using MyoVIEW software (DMT). The arteriole was continuously superfused with the Krebs buffer bubbled with a gas mixture of 21% $O_2$, 5% $CO_2$, and 74% $N_2$. The arteriole was pressurized at an intraluminal pressure of 8 mmHg by a DMT pressure regulator. The chamber temperature was kept at 37°C by a DMT culture myograph heat controller. After 4 h of incubation, the oligonucleotide in the lumen was gradually washed out by generating a flow of the normal Krebs buffer through the lumen. Endothelium-dependent vasodilation was analyzed 24 h later using the DMT system with increasing doses of acetylcholine and L-NAME.

### Development of miR-29b1 mutant rats

The *Mir29b-1* gene was targeted for mutation using custom-made Transcriptional Activator-Like Effector Nucleases (TALENs, Cellectis Bioresearch) which targeted the sequence TTTAAATAGT**GATTGTC**tagcaccatttgaaaTCAGTGTTCTTGGTGGA where each TALEN monomer binds the target sequences underlined on opposite strands, separated by a spacer (lowercase). The mature sequence for rno-miR-29b-3p is shown in bold (miRBase accession # MIMAT0000801). *In vitro*-transcribed mRNAs encoding the *Mir29b-1* TALENs were injected into one-cell SS-Chr 13BN/Mcwi (SS-Chr13[BN]; RGD ID: 629523) rat embryos as described previously (Hakamata & Kobayashi, 2010). SS-Chr13[BN] rats are inbred rats that have been used as normal control in studies of vascular dysfunction (Cowley *et al*, 2001; Drenjancevic-Peric *et al*, 2003). A mutant rat line, SS-Chr 13BN-*Mir29b1*[em1Mcwi], hereafter referred to as *Mir29b-1/a* mutant or *Mir29b-1/a*[−/−] rat, was generated, having a TALEN-induced 4-bp deletion within the TALEN target spacer, TTTAAATAGT**GATTGTC**tagca—ttgaaaTCAGTGTTCTTGGTGGA confirmed by Sanger sequencing and predicted to disrupt the rno-miR-29b-3p sequence. Heterozygous breeder pairs were set up, giving rise to litters containing wild-type (WT), heterozygous (Het), and homozygous mutant rats for phenotyping. Genotyping was performed using a fragment analysis protocol, using M13-miR29b1_F (5′-*TGTAAAACGACGGCCAGT* AATGCAGCAAGTGACT GACATGTC-3′) and mir29b1_R (5′-GGGCCTTCTGTCTGTTGTACA TG-3′) primer sequences, where the M13 forward tag is in italics, and products analyzed using a simple sequence length polymorphism fluorescent genotyping assay using an ABI 3730xl DNA analyzer as described (Moreno *et al*, 2005).

### Statistical analyses

Data were analyzed by chi-squared tests, unpaired *t*-tests, Wilcoxon rank-sum test, or two-way ANOVA followed by Holm–Sidak method, as indicated in the text. Data were reported as mean $\pm$ SEM with *n* values indicating biological replicates. Exact *P*-values for all comparisons are available in Appendix Table S4.

### Other methods

Methods for measuring NO levels in vessels and cells, small RNA and RNA deep sequencing and data analysis, real-time PCR, rat blood pressure measurements, isolation of rat arterioles for vascular function measurements, extraction of endothelium-enriched fractions from rat vessels, cell culture, Western blotting, and 3′ UTR reporter assay are included in the Appendix.

**Expanded View** for this article is available online.

## Acknowledgements

This work was supported by US National Institutes of Health grants HL125409 (MEW and ML), DK076169 sub-award 25732 (MEW and ML), HL121233 (ML), HL082798-6186 (ML), K23HL089326 (MEW), and HL128240 (MEW), American Heart Association grant 15SFRN23910002 (ML), and Advancing a Healthier Wisconsin Endowment. DMJ is a recipient of a National Research Service Award Training Program of UL1TR001436 and 1TL1TR001437.

## Author contributions

MEW and ML conceived, designed, and led the study. DMJ performed most of the experiments involving rats and cultured cells and some expression analysis of human arterioles. JW performed human vascular function experiments. YL performed several molecular experiments. AMG led the development of the mutant rat model. AJK performed expression analysis and assisted with microRNA manipulation experiments. PL led the analysis of deep sequencing data. RY performed arteriole isolation. GZ contributed to the molecular and blood pressure analysis of the mutant rat. MC contributed to the deep sequencing experiment. CC contributed to the analysis of deep sequencing data. MEW, MM, AB, and ST carried out human participant recruitment, clinical phenotyping, and biopsy. MJT contributed to molecular and cellular experiments. KU contributed to the blood pressure analysis of the mutant rat. ML and MEW drafted the manuscript with assistance from DMJ and other co-authors.

## Conflict of interest

Dr. Widlansky receives investigator-initiated grant funding from Merck Sharp & Dohme Corp. and Medtronic for work unrelated to this manuscript.

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
