## [Review Process File · EMBO Molecular Medicine]

mir-29 Contributes to Normal Endothelial Function and can Restore it in Cardiometabolic Disorders

Michael E. Widlansky, David M. Jensen, Jingli Wan, Yong Liu, Aron M. Geurts, Alison J. Kriegel, Pengyuan Liu, Rong Ying, Guangyuan Zhang, Marc Casati, Chen Chu, Mobin Malik, Amberly Branum, Michael J. Tanner, Sudhi Tyagi, Kristie Usa, Mingyu Liang

Review timeline:

Submission date:	17 May 2017
Editorial Decision:	22 June 2017
Revision received:	18 November 2017
Editorial Decision:	11 December 2017
Revision received:	29 December 2017
Accepted:	08 January 2018

Editors: Roberto Buccione and Céline Carret

Transaction Report:

1st Editorial Decision

22 June 2017

Thank you for the submission of your manuscript to EMBO Molecular Medicine and many apologies for the unusual delay in providing you with a decision, due to difficulties in recruiting appropriate reviewers and then obtaining their evaluations in a timely fashion.

We have now heard back from the three Reviewers whom we asked to evaluate your manuscript

As you will see, the reviewers offer strikingly overlapping evaluations. Although there is clear appreciation for the potential interest of your work and the value of the animal model used, they collectively raise a number of important concerns, which I would basically summarise as follows: 1) lack of important controls; 2) missing link between miR-29 and Lypla1, 3) confusion (in nomenclature and experimental) concerning the miR-29 subtypes; 4) unclear evidence of the role of NO, lack of validation in the human arterioles. The reviewers also list a number of additional items of concern.

After our reviewer cross-commenting exercise, there was agreement that you should convincingly show that NO and the endothelial miR-29 is likely to cause the phenotype *in vivo*, and also demonstrate the causal link between miR-29-Lypla1-eNOS. It was also mentioned that in principle this could be achieved without an endothelial-specific animal model, which would perhaps take too long to pursue, unless of course you have this model available.

In conclusion, while publication of the paper cannot be considered at this stage, given the potential interest of your findings and after internal discussion, we have decided to give you the opportunity to address the criticisms. Please consider that the concerns raised are of great importance for us as they impinge on the overall quality and robustness of experimental support for the main conclusions.

We are thus prepared to consider a substantially revised submission, with the understanding that the reviewers' concerns as indicated above must be addressed with additional experimental data, where appropriate and that acceptance of the manuscript will entail a second round of review.

Since the required revision in this case appears to require a significant amount of time, additional work and experimentation and might be technically challenging, I would therefore understand if you chose to rather seek publication elsewhere at this stage. Should you do so, we would welcome a message to this effect. Please note that it is EMBO Molecular Medicine policy to allow a single round of revision only and that, therefore, acceptance or rejection of the manuscript will depend on the completeness of your responses included in the next, final version of the manuscript. Indeed, I would strongly advise against submitting a partially revised manuscript.

I look forward to reading your revised manuscript in due time and would appreciate a message concerning your interest (or not) in pursuing a revision.

***** Reviewer's comments *****

Referee #1 (Comments on Novelty/Model System):

The TALEN rat is an excellent model system. The ex vivo vessel studies are current technology. The only area that could be improved would be a tissue specific transgenic mouse with expression in endothelium only.

Referee #1 (Remarks):

General Comments: In this paper, the authors investigated the role of microRNAs (miRNA) in endothelial dysfunction in the setting of cardiometabolic disorders represented by type 2 diabetes (T2DM). Their hypothesis is there are miRNA that interfere with normal endothelium function. By using various approaches, they show that miR-29 is required for normal endothelial function and can restore endothelium-dependent vasodilation in resistance arterioles from T2DM patients by promoting nitric oxide (NO) production. To prove their hypothesis, the authors obtained arterioles from T2DM patients. The arterioles show impaired endothelium-dependent vasodilation. Intraluminal delivery of miR-29a-3p or miR-29b-3p mimics restored normal endothelium-dependent vasodilation in T2DM arterioles. Intraluminal delivery of anti-miR-29b-3p in arterioles from non-DM human subjects or targeted mutation of *Mir29b-1* gene in rats led to impaired endothelium-dependent vasodilation and exacerbation of hypertension in the rats. The results are novel and the techniques are very contemporary. The experiments are well designed and the data are of high quality. However, there are a number of issues that decrease the scientific importance.

Major comments:

1. The description of the miR29 subtypes is confusing and difficult to follow. Several experiments in Fig1 and 2 used anti-miR-29b-3p and a miR-29b mimic. Although the authors showed miR-29a-3p was increased in human arteriole sample, they failed to show the increase of miR-29b-3p in human arteriole sample using real-time PCR.
2. In Fig2, the authors claim that miR-29 is required for normal endothelial function but almost no data on miR-29a was shown. For example, does intraluminal delivery of anti-miR-29a-3p impair vasodilation?
3. The decrease of miR-29a-3p in *Mir2b-1* mutant rat doesn't suggest the causal relationship of miR-29a-3p necessary for normal endothelial function. The lack of correlation between the expression level and phenotype is not well explained.
4. Does the *Mir2b-1* mutant rat have phenotypes in resistance vessel rich organs such as lung, kidney and brain? The authors mentioned miR-29 contributing to pulmonary arterial hypertension, which is also a resistance vessel dysfunction disease. The role of miR-29 here then seems contradictory in terms of metabolism and vessel function.
5. In hMVEC, does anti-miR29b treatment lead to up-regulation of *Lypla1* and decrease of eNOS activity? What happens to eNOS activity when there is both anti-miR29b and *Lypla1* siRNA? In Fig3B, the authors show anti-miR29b reduces NO, then how is this achieved? Is it through *Lypla1*? The authors claim that *Lypla1* only functions when it was suppressed by miR-29b-3p, but there should also be discussion about the mechanism why NO goes down with anti-miR29b.

6. Do protein and mRNA levels of Lypla1 and Gap43 go up in human arteriole sample?

The authors claim that up-regulated miR-29 in T2DM arterioles is a failed attempt of compensation, could this be due to its loss of control of downstream targets? There is no data to support "the failure of compensation". It would be important to know what happened to the Lypla1 in human sample. The argument would be strengthened by data of RT-PCR quantification of Lypla1 and Gap43 in human sample.

Referee #2 (Remarks):

The manuscript by Widlansky and colleagues investigates the role of microRNA29 in endothelial cell functions in Type 2 diabetes. It is a three pronged approach, investigating arterioles taken from human patients, in vitro experiments and also investigations in a newly established animal model of mutant MiR-29.

Their conclusions are that miR-29 acts through its target gene Lypla1 to regulate NO production which impacts on endothelial cell function, namely endothelial-dependent vasodilation.

There are a large number of major issues.

1. The authors fail to establish that Lypla1 is the critical miR-29 target that operates in the human or rat setting. They have the set-up to do it. Is Lypla1 regulated in the T2DM samples and especially in the endothelium? If you overexpress the mimic in the arterioles where there is a subsequent rescue of vasodilation, then Lypla1 levels should be changed in the vessels. Transfection of the siRNA Lypla1 should rescue.

2. Figure1B shows the results for the antagomir. But there is no control in this experiment with normal arterioles, performed at the same time under the same experimental conditions. The fact that the normals are in Figure2, which is where the rat model is discussed, suggests these normals were NOT performed at the same time as Figure1. Further, it is important to show that the antagomir decreases miR-29 and if so does it work on both miR-29a and miR-29b in these arterioles.

3. Is miR 29b statistically different in response to miR-29b? It looks like it maybe only at the high dose of Ach. Not at the lower doses. What does this then tell you?

4. Do miR-29a and miR-29b both target Lypla1? There is some precedence in the literature, especially in endothelial cells that there can be preferential target selection with the different isoforms of microRNAs. This may explain why miR-29b works better than miR-29a in this assay even though there is considerably more miR-29a in the arterioles.

5. The rat model, which the authors have developed, is a very interesting model. Does siRNA to Lypla1, in the rat arterioles rescue the vasodilation problems? If mimics can be transfected then siRNAs should also work.

6. In Figure 1E, does the antagomir reduce the dilation of the WT and does this regulate Lypla1 levels. Again this is an important control to verify the system.

7. In Figure3, the authors show data on effects of L-NAME. However, they fail to make the direct link from miR-29 and Lypla1 and NO. As they point out in the discussion, there are many NO regulators that are altered (see Fig4A) so why is Lypla1 the major miR-29 target to affect NO? In Figure3E there are no groups without L-NAME-this is essential.

8. SupplFigs2 shows uptake of the microRNAs. So it looks like miR-29a upregulates miR-29b but not the reverse. Given that miR-29b is very low under normal circumstances this 2fold change is highly significant and impacts on the other results. Do the authors see the same regulation when they use normal human arterioles? There is no Figure 1E (as stated in the legend to this Fig).

9. The hypothesis that in the human T2DM, the miR-29 is a mutant form and thus fails to operate is an interesting possibility but without some evidence for this, then the MS lacks a clear take home message. This is especially also given the fact that they also fail to make the miR-29-Lypla1-NO connection. Does the Mir29 in 2TDM get processed differently? Does overexpression of the miRs in normal arterioles result in similar levels as in the 2TDM samples? Is it possible to sequence the miR-29 that is in the plasma in the T2DM? If it is not a mutant form, then where is the problem in the T2DM? This does not come across in the Discussion, which is often vague (eg 1st paragraph p19, what are the experiments that could have alternate interpretations) and sometimes contains information that is not really relevant (eg top page 17).

Referee #3 (Remarks):

In this manuscript by Widlansky and colleagues, the authors describe that miR-29 is required for normal endothelial-dependent dilatatory function. The KO rat displays hypertension and the mechanism the authors describe relates to NO production by ECs. The work is interesting, but several caveats need to be addressed:

Major comments:

1. Figure 2C: What is the expression of miR-29a, b and c in these fractions? Does miR-29 correlate with PECAM1? Is miR-29 higher in EC elute than in the remainder?
2. The authors use the nomenclature miR-29b-1-/- for the KO rat, but in fact it should be termed miR-29b-1;miR-29a-/- . What happens to miR-29b-1-5p and miR-29a-5p?
3. Figure 4B: This is not an unbiased approach to assess which genes/pathways are regulated by miR-29. There are several (publicly available) bioinformatics tools to identify pathways that are regulated in a more unbiased manner. Does the NO synthesis pathway then also stand out?
4. Does miR-29 affect palmitoylation of eNOS?
5. Figure 2A and 3D: L-NAME completely blocks dilation, but this is not direct proof that miR-29 regulates dilation via NO. Does addition of an NO donor normalize the effects of miR-29 blockade as seen in figure 2A? Do siRNAs against LYPLA1 normalize this dilation as well?
6. MiR-29 was previously shown to repress fibrosis (van Rooji et al, PNAS 2008) and induce aneurysms (Maegdefessel et al, JCI 2012). These processes are linked to fibroblast/SMC biology and affect ECM and stiffness. Therefore effects on vascular SMCs and vessel stiffness are expected in the total KO of miR-29 in rats. Unless I under-interpret some of the experiments, there is no conclusive proof that this is not the case. Can a pharmacological increase of endothelial function or NO donors induce relaxation/dilation of vessel in the miR-29 KO rat?

Minor comment:

1. Figure 4C and E: n=2-3 and n=2. This is insufficient to draw conclusions (or perform statistics from). Please increase the number of experiments.

1st Revision - authors' response

18 November 2017

Editor's comments

As you will see the reviewers offer strikingly overlapping evaluations. Although there is clear appreciation for the potential interest of your work and the value of the animal model used, they collectively raise a number of important concerns, which I would basically summarise as follows: 1) lack of important controls; 2) missing link between miR-29 and Lypla1, 3) confusion (in nomenclature and experimental) concerning the miR-29 subtypes; 4) unclear evidence of the role of NO, lack of validation in the human arterioles. The reviewers also list a number of additional items of concern.

After our reviewer cross-commenting exercise, there was agreement that you should convincingly show that NO and the endothelial miR-29 is likely to cause the phenotype in vivo, and also demonstrate the causal link between miR-29-Lypla1-eNOS. It was also mentioned that in principle this could be achieved without an endothelial-specific animal model, which would perhaps take too long to pursue, unless of course you have this model available.

RESPONSE: We have added approximately 10 sets of new experiments, including recruitment, biopsy and study of an additional 17 human subjects and examination of the effect of Lypla1 siRNA on endothelium-dependent vasodilation in the settings of T2DM or miR-29 suppression or mutation. The results strongly support the conclusions that miR-29 promotes endothelium-dependent vasodilation via NO and that Lypla1 contributes to these effects of miR-29.

***** Reviewer's comments *****

Referee #1 (Comments on Novelty/Model System):

The TALEN rat is an excellent model system. The ex vivo vessel studies are current technology. The only area that could be improved would be a tissue specific transgenic mouse with expression in endothelium only.

RESPONSE: An endothelium-specific transgenic model would be great. As the editor and reviewers agreed on, such a model would take a long time to pursue and the main concerns about the paper could be addressed without such a model. We have obtained a larger amount of new data that we believe have addressed the reviewers' main concerns as detailed below.

Referee #1 (Remarks):

General Comments: The results are novel and the techniques are very contemporary. The experiments are well designed and the data are of high quality.

RESPONSE: Thank you very much for appreciating the novelty and rigor of our study!

Major comments:

1. The description of the miR29 subtypes is confusing and difficult to follow. Several experiments in Fig1 and 2 used anti-miR-29b-3p and a miR-29b mimic. Although the authors showed miR-29a-3p was increased in human arteriole sample, they failed to show the increase of miR-29b-3p in human arteriole sample using real-time PCR.

RESPONSE: We have measured miR-29b-3p using real-time PCR and confirmed it is increased in human T2DM arterioles (see revised Supplemental Figure S1). We have made sure nomenclatures are consistent throughout. Some of the nomenclatures in the figures are shortened to save space, but they are spelled out in full in figure legends and the text.

2. In Fig2, the authors claim that miR-29 is required for normal endothelial function but almost no data on miR-29a was shown. For example, does intraluminal delivery of anti-miR-29a-3p impair vasodilation?

RESPONSE: We have performed intraluminal delivery of anti-miR-29a-3p and anti-miR-29b-3p in normal rat gluteal arterioles. Both anti-miR's impaired Ach-induced vasodilation (new Figure 2B).

3. The decrease of miR-29a-3p in Mir2b-1 mutant rat doesn't suggest the causal relationship of miR-29a-3p necessary for normal endothelial function. The lack of correlation between the expression level and phenotype is not well explained.

RESPONSE: We believe data from #2 above would help address this concern.

4. Does the Mir2b-1 mutant rat have phenotypes in resistance vessel rich organs such as lung, kidney and brain? The authors mentioned miR-29 contributing to pulmonary arterial hypertension, which is also a resistance vessel dysfunction disease. The role of miR-29 here then seems contradictory in terms of metabolism and vessel function.

RESPONSE: It would be interesting to examine lung vessels in the mutant rat. We have not done that.

5. In hMVEC, does anti-miR29b treatment lead to up-regulation of Lypla1 and decrease of eNOS activity? What happens to eNOS activity when there is both anti-miR29b and Lypla1 siRNA? In Fig3B, the authors show anti-miR29b reduces NO, then how is this achieved? Is it through Lypla1? The authors claim that Lypla1 only functions when it was suppressed by miR-29b-3p, but there should also be discussion about the mechanism why NO goes down with anti-miR29b.

RESPONSE: We have performed all of the experiments the reviewer suggested. Following transfection with anti-miR-29b, Lypla1 mRNA abundance was increased (new Figure 5E) and eNOS activity, measured as A23187-induced and L-NAME-inhibitable NO generation, was decreased (new Figure 5F). Additional transfection with Lypla1 siRNA increased NO generation in cells transfected with anti-miR-29b (new Figure 5F).

6. Do protein and mRNA levels of Lypla1 and Gap43 go up in human arteriole sample? The authors claim that up-regulated miR-29 in T2DM arterioles is a failed attempt of compensation, could this be due to its loss of control of downstream targets? There is no data to support "the failure of compensation". It would be important to know what happened to the Lypla1 in human sample. The

argument would be strengthened by data of RT-PCR quantification of Lypla1 and Gap43 in human sample.

RESPONSE: We had remaining RNA samples available for 30 of the 38 original patients (17 non-DM and 13 DM). We have measured Lypla1 and Gap43 mRNA in these samples. LYPLA1 mRNA abundance was significantly higher in the DM arterioles compared to non-DM arterioles (new Figure 5H). GAP43, while trending towards higher abundance in the microvessels of the DM patients, did not reach statistical significance. We do not have enough human arteriole samples for protein analysis.

Referee #2 (Remarks):

1. The authors fail to establish that Lypla1 is the critical miR-29 target that operates in the human or rat setting. They have the set-up to do it. Is Lypla1 regulated in the T2DM samples and especially in the endothelium? If you overexpress the mimic in the arterioles where there is a subsequent rescue of vasodilation, then Lypla1 levels should be changed in the vessels. Transfection of the siRNA Lypla1 should rescue.

RESPONSE: We have performed the experiments the reviewer suggested. Lypla1 is up-regulated in human T2DM arterioles (new Figure 5H), although we cannot separate the endothelium from these arterioles for analysis. Transfection of human T2DM arterioles with Lypla1 siRNA significantly improved vasodilation (new Figure 5I). In normal human arterioles, transfection of anti-miR-29 impaired vasodilation, which was rescued by additional transfection with Lypla1 siRNA (new Figure 5J).

2. Figure 1B shows the results for the antagomir. But there is no control in this experiment with normal arterioles, performed at the same time under the same experimental conditions. The fact that the normals are in Figure 2, which is where the rat model is discussed, suggests these normals were NOT performed at the same time as Figure 1. Further, it is important to show that the antagomir decreases miR-29 and if so does it work on both miR-29a and miR-29b in these arterioles.

RESPONSE: Fig 1B and Fig 2A were performed during the same time period under the same experimental conditions. We put Fig 2A in Fig 2 to show results from experiments manipulating the "normal" together. We have clarified this in the figure legend. We do not think measurement of miR-29 following anti-miR-29b is essential or would justify the amount of work (subject recruitment, biopsy and intraluminal transfection) required. Besides, anti-miR could block miR function without degrading the miR. Nevertheless, we have added an anti-miR-29a experiment in rat arterioles (new Figure 2B) in response to other comments from reviewers, which should help to address the reviewer's concern here to some extent.

3. Is miR 29b statistically different in response to miR-29b? It looks like it maybe only at the high dose of Ach. Not at the lower doses. What does this then tell you?

RESPONSE: We think the reviewer was asking about the improvement of dilation by miR-29a and miR-29b shown in Figure 1C and 1D. The responses to the two miR's were not statistically different ($p=0.57$ at the highest dose of Ach comparing miR-29a to miR-29b).

4. Do miR-29a and miR-29b both target Lypla1? There is some precedence in the literature, especially in endothelial cells that there can be preferential target selection with the different isoforms of microRNAs. This may explain why miR-29b works better than miR-29a in this assay even though there is considerably more miR-29a in the arterioles.

RESPONSE: It is most likely that miR-29a and miR-29b both target Lypla1, although the possibility of different degrees of targeting is very interesting and would require further investigation.

5. The rat model, which the authors have developed, is a very interesting model. Does siRNA to Lypla1, in the rat arterioles rescue the vasodilation problems? If mimics can be transfected then siRNAs should also work.

RESPONSE: We have performed the suggested experiment. Lypla1 siRNA significantly improved Ach-induced dilation of arterioles from mutant rats (new Figure 5G).

6. In Figure 1E, does the antagomir reduce the dilation of the WT and does this regulate Lypla1 levels. Again this is an important control to verify the system.

RESPONSE: We think the reviewer was referring to Fig 2E (now Fig 2F). We have performed the suggested experiments. Transfection with anti-miR-29a-3p or anti-miR-29b-3p significantly impaired Ach-induced dilation in WT rats (new Figure 2B). While we did not measure Lypla1 in these vessels, we have added several other experiments to support the role of Lypla1 (see new Figure 5).

7. In Figure3, the authors show data on effects of L-NAME. However, they fail to make the direct link from miR-29 and Lypla1 and NO. As they point out in the discussion, there are many NO regulators that are altered (see Fig4A) so why is Lypla1 the major miR-29 target to affect NO? In Figure3E there are no groups without L-NAME-this is essential.

RESPONSE: Lypla1 has a clearly discernable role, but is probably not the only mediator. As indicated throughout the response to reviewer comments, we have added several new experiments in human and rat arterioles using Lypla1 siRNA, which support Lypla1 being one of the important links between miR-29 and NO (see new Figure 5). Groups without L-NAME for Fig 3E are shown in Fig 2F so that all L-NAME experiments are presented together in Fig 3. The experiments with and without L-NAME were done in parallel and under otherwise similar conditions. We have clarified this in the figure legend.

8. SupplFigs2 shows uptake of the microRNAs. So it looks like miR-29a upregulates miR-29b but not the reverse. Given that miR-29b is very low under normal circumstances this 2fold change is highly significant and impacts on the other results. Do the authors see the same regulation when they use normal human arterioles? There is no Figure 1E (as stated in the legend to this Fig).

RESPONSE: We apologize that the labels in the figure might have been confusing. 29a mimic doesn't up-regulate 29b. Also, we should have referred to Fig 1C and 1D instead of 1D and 1E. We apologize for the error.

9. The hypothesis that in the human T2DM, the miR-29 is a mutant form and thus fails to operate is an interesting possibility but without some evidence for this, then the MS lacks a clear take home message. This is especially also given the fact that they also fail to make the miR-29-Lypla1-NO connection. Does the Mir29 in 2TDM get processed differently? Does overexpression of the miRs in normal arterioles result in similar levels as in the 2TDM samples? Is it possible to sequence the miR-29 that is in the plasma in the T2DM? If it is not a mutant form, then where is the problem in the T2DM? This does not come across in the Discussion, which is often vague (eg 1st paragraph p19, what are the experiments that could have alternate interpretations) and sometimes contains information that is not really relevant (eg top page 17).

RESPONSE: The strong take-home message from the current study is miR-29 is required for endothelium-dependent vasodilation in normal arterioles in human and rat and can restore endothelium-dependent vasodilation in human T2DM arterioles. The new experiments we have added in the revision have provided further support for the miR-29-Lypla1-NO connection. The question of why endogenous miR-29 lost its function in T2DM arterioles requires an extensive series of studies to answer, which we are currently working on. It is beyond the scope of the current manuscript.

Referee #3 (Remarks):

In this manuscript by Widlansky and colleagues, the authors describe that miR-29 is required for normal endothelial-dependent dilatory function. The KO rat displays hypertension and the mechanism the authors describe relates to NO production by ECs. The work is interesting, but several caveats need to be addressed:

Major comments:

1. Figure 2C: What is the expression of miR-29a, b and c in these fractions? Does miR-29 correlate with PECAM1? Is miR-29 higher in EC elute than in the remainder?

RESPONSE: We have measured miR-29a-3p, 29a-5p, 29b-3p, 29b1-5p, and 29c-3p in these gluteal arteriole fractions using qPCR (described on pages 11-12 in addition to Figure 2E). miR-29a-5p and miR-29b1-5p (the passenger strands) were not detectable in any group. miR-29a-3p and miR-29b-3p were reduced in heterozygotes and further reduced in homozygote rats, matching the data already included in the original manuscript (Fig 2E). miR-29a-3p and miR-29b-3p were not significantly different between the EC elute fraction and the vessel remainder for any genotype of the Mir29 mutant rats. miR-29c was not significantly different between genotypes or between the EC elute fraction and vessel remainder.

2. The authors use the nomenclature miR-29b-1/- for the KO rat, but in fact it should be termed miR-29b-1;miR-29a-/- . What happens to miR-29b-1-5p and miR-29a-5p?

RESPONSE: Good suggestion. We have changed the terminology to Mir29b-1/a -/-. Neither of the 5p strands was detectable as indicated in the response to the last comment.

3. Figure 4B: This is not an unbiased approach to assess which genes/pathways are regulated by miR-29. There are several (publicly available) bioinformatics tools to identify pathways that are regulated in a more unbiased manner. Does the NO synthesis pathway then also stand out?

RESPONSE: We have the "unbiased" pathway analysis shown in Supplemental Dataset 4, which shows several vascular pathways. NO is not in Supplemental Dataset 4 directly, but that might be due to the granularity of Gene Ontology. Fig 4B is a hypothesis-driven analysis, and it complements the unbiased analysis.

4. Does miR-29 affect palmitoylation of eNOS?

RESPONSE: We attempted to measure palmitoylation of eNOS following transfection of miR-29b mimic in HMVEC-d cells following the protocol reported by Forrester et al (J of Lipid Research 2011) but were unsuccessful. The samples which should have represented all palmitoylated proteins in the cells did not show any protein present after coomassie staining. We suspect the purification reagent was not efficient in pulling out any palmitoylated protein. We do have strong data showing miR-29 promotes eNOS activity by targeting Lypla1 (Fig 4 and new Fig 5). The only known connection between Lypla1 and eNOS is palmitoylation.

5. Figure 2A and 3D: L-NAME completely blocks dilation, but this is not direct proof that miR-29 regulates dilation via NO. Does addition of an NO donor normalize the effects of miR-29 blockade as seen in figure 2A? Do siRNAs against LYPLA1 normalize this dilation as well?

RESPONSE: We have performed the suggested experiments. An NO donor (spermine NONOate) normalized the effects of miR-29 blockade seen in figure 2A (new Supplemental Figure S3). Lypla1 siRNA also rescued Ach-induced dilation in arterioles from T2DM patients (new Figure 5I) or arterioles from non-DM subject with miR-29 blockade (new Figure 5J).

6. MiR-29 was previously shown to repress fibrosis (van Rooji et al, PNAS 2008) and induce aneurysms (Maegdefessel et al, JCI 2012). These processes are linked to fibroblast/SMC biology and affect ECM and stiffness. Therefore effects on vascular SMCs and vessel stiffness are expected in the total KO of miR-29 in rats. Unless I under-interpret some of the experiments, there is no conclusive proof that this is not the case. Can a pharmacological increase of endothelial function or NO donors induce relaxation/dilation of vessel in the miR-29 KO rat?

RESPONSE: We have measured dilation of isolated gluteal arterioles from homozygous *Mir29b-1/a* mutant rats with spermine NONOate (new Supplemental Figure S5). The vessels dilated to a max of ~70%, compared with the ~30% max dilation in response to Ach, indicating that the dilatory response to NO donors remains intact in the mutant rat.

Minor comment:

1. Figure 4C and E: n=2-3 and n=2. This is insufficient to draw conclusions (or perform statistics from). Please increase the number of experiments.

RESPONSE: We have increased the n to 4 for the siRNA experiment (now Figure 5A). The effect was clear-cut.

2nd Editorial Decision

11 December 2017

Thank you for the submission of your revised manuscript to EMBO Molecular Medicine. We have now received the enclosed reports from the referees that were asked to re-assess it. As you will see the reviewers are now globally supportive and I am pleased to inform you that we will be able to accept your manuscript pending a few final editorial amendments.

***** Reviewer's comments *****

Referee #1 (Remarks for Author):

I think they have successfully addressed almost all the concerns. The only thing I feel this paper lacking of is the discussion about the mechanism of why this protective mir29 is actually increased in diabetes patients (whether through increased lylpa-1 or other pathways) and if this increased expression of protective miRNA is a common phenomenon in diabetes.

Referee #2 (Remarks for Author):

This is a much-improved MS and clear in the message.

Referee #3 (Remarks for Author):

The authors addressed my concerns.

Corresponding Author Name: Michael Widlansky and Mingyu Liang

Manuscript Number: EMM-2017-08046